# Long-Term Outcomes of 5-Fluorouracil-Related Early-Onset Toxicities: A Retrospective Cohort Study

**DOI:** 10.3390/cancers16234050

**Published:** 2024-12-03

**Authors:** Nicolás Tentoni, Ryan Combs, Miriam Hwang, Suzanne Ward, Andrea McCracken, Jennifer Lowe, Scott C. Howard

**Affiliations:** 1Resonance, Memphis, TN 38104, USA; ryan.combs@resonancehealth.org (R.C.); miriam.hwang@resonancehealth.org (M.H.); jennifer.lowe@resonancehealth.org (J.L.); scott.howard@resonancehealth.org (S.C.H.); 2Laboratory of Applied Statistics in the Health Sciences, Faculty of Medicine, University of Buenos Aires, Buenos Aires C1121 ABG, Argentina; 3BTG International Inc., West Conshohocken, PA 19428, USA; suzanne.ward@serb.com; 4Guardian Research Network, Spartanburg, SC 29303, USA; amccracken@guardianresearch.org; 5Sant Joan de Déu Hospital Barcelona, 08950 Barcelona, Spain; 6Yeolyan National Hematology Center, Yerevan 0014, Armenia

**Keywords:** chemotherapy adverse events, 5-fluorouracil toxicity, prognostic factors, clinical outcomes, early-onset toxicities

## Abstract

In this retrospective cohort study of 3988 treatment-naive cancer patients, the median overall survival was 2.5 years in those who experienced early-onset toxicity from 5-FU during their first FOLFOX/FOLFIRINOX cycle, compared with 5.3 years in those who did not, a significant difference. Early-onset toxicity from 5-FU is a significant prognostic factor for survival in treatment-naive patients undergoing their first cycle of FOLFOX or FOLFIRINOX.

## 1. Introduction

5-fluorouracil (5-FU) is a key chemotherapeutic agent in regimens used to treat various solid tumors. Its active metabolite—fluoro-deoxyuridine monophosphate (FdUMP)—inhibits thymidylate synthase and subsequent DNA synthesis in tumor cells and blocks protein synthesis by incorporating into RNA, ultimately leading to cell death [1,2]. The FOLFOX regimen, composed of folinic acid, 5-FU, and oxaliplatin, is the cornerstone of therapy for advanced and metastatic colorectal cancer [3]. It is also standard of care for the second-line treatment of advanced biliary tract cancers and a feasible treatment for pancreatic cancer [4,5]. The FOLFIRINOX regimen incorporates the same agents as FOLFOX plus the topoisomerase inhibitor irinotecan; it is the main treatment for metastatic pancreatic cancer and advanced colorectal cancer [6,7,8,9]. However, treatment with 5-FU can be complicated by various toxicities that can lead to dose reduction or treatment cessation. Gastrointestinal toxicities and myelosuppression are most common, but cardiac (e.g., vasospasm, heart failure, myocardial infarction) and neurologic (e.g., acute cerebellar syndrome, encephalopathy) toxicities may also occur [1,10]. These toxicities can be particularly devastating in patients with deficiencies or functional variations in enzymes such as dihydropyrimidine dehydrogenase (DPD), thymidylate synthase (TYMS), and orotate phosphoribosyl transferase (ORPT), which can lead to the accumulation of 5-FU and an increased sensitivity to 5-FU and its metabolites [1,11,12,13].

Severe life-threatening toxicity is reported in up to 30% of patients receiving 5-FU and mortality due to 5-FU toxicity is approximately 1%, both closely associated with DPD deficiency [14,15]. Early-onset severe toxicities can occur during or following the first course of 5-FU treatment, thus rendering rapid recognition and treatment essential to preserve the patient’s well-being [16,17]. Uridine triacetate is the only antidote for fluoropyrimidine approved by the Food and Drug Administration (FDA) for both overdose and severe early-onset toxicity; it has shown efficacy in reversing toxicity and allowing the resumption of chemotherapy when provided within 96 h of the cessation of fluoropyrimidine delivery [18]. It is administered orally and well tolerated in both adult and pediatric patients with mild and infrequent adverse reactions reported (e.g., nausea, vomiting, diarrhea). As genetic testing for DPD deficiency is currently not universally conducted prior to fluoropyrimidine treatment, the availability of uridine triacetate can be a source of effective supportive therapy in the event of an overdose or severe toxicity.

Data on 5-FU-related toxicities are generally reported from clinical trials, the majority of which enroll highly selected patient populations and take place under controlled settings in academic health centers that often have more supportive therapy options, including uridine triacetate [19,20]. The incidence, severity, and clinical impact of 5-FU toxicities have not been well documented in community oncology practices, where 85% of oncology patients in the United States receive their care. Patients in community practices comprise diverse socioeconomics, races, and ethnicities and present with various comorbidities, all of which may be associated with unusual patterns of toxicities and management, and thus provide real-world data on treatment outcomes [19,21,22]. This study was undertaken to investigate the association between 5-FU-related early-onset toxicities and long-term survival outcomes in cancer patients in community oncology practices. Specifically, to ensure toxicities were attributable to 5-FU, this study focused solely on antimetabolite-naïve patients who had received their first cycle of first-line FOLFOX or FOLFIRINOX therapy.

## 2. Materials and Methods

### 2.1. Study Design and Setting

This was a multicenter observational cohort study using information from the Guardian Research Network (GRN). The GRN is a healthcare network of 70 community oncology practices spanning 14 states in the central and southern geographic locations of the United States (Appendix A). The practices within the GRN (hospitals, outpatient clinics, surgical centers, and infusion centers, etc.) pool their data into a shared anonymized data warehouse using an electronic medical records (EMR) system to include all ICD codes, medications, diagnostic results (e.g., laboratory values, imaging, and pathology reports), therapeutic procedures, clinical notes, and toxicities.

### 2.2. Patient Population

Patients diagnosed with cancer (based on ICD-10 codes) who received their first dose of 5-FU from 1 January 2015 through 1 August 2023 were eligible for inclusion. Patients were excluded if their 5-FU administration was not a component of either the FOLFOX or FOLFIRINOX regimen or if they had previous exposure to 5-FU or capecitabine treatment prior to the index cancer diagnosis.

### 2.3. Explanatory Variable

The occurrence of an early-onset 5-FU-related toxicity in the first cycle of FOLFOX or FOLFIRINOX was the main explanatory variable. Toxicities were identified and categorized as gastrointestinal (vomiting, diarrhea, mucositis), hematopoietic (thrombocytopenia, neutropenia), cardiovascular, and neurological using specific criteria for each based on ICD-10 codes, laboratory results, and procedures that were recorded in the EMR (Appendix A). “Early-onset” was defined as having occurred during the 5-FU infusion or up to 96 h after infusion completion.

### 2.4. Endpoints

The primary endpoint was overall survival and secondary endpoints included early treatment cessation and early hospital admission. Early treatment cessation was defined as having less than 8 cycles of FOLFOX or FOLFIRINOX in the 8 months following the first cycle. The cut-off of less than 8 completed cycles for early cessation was based on a previous study that reported 8 out of 12 cycles of adjuvant FOLFOX was associated with significant overall survival [23]. Early hospital admission was defined as an unplanned admission that occurred during the 5-FU infusion or up to 96 h after infusion completion of the first cycle of FOLFOX or FOLFIRINOX.

### 2.5. Diagnosis, Baseline Characteristics, and Potential Confounders

Cancer diagnosis and baseline characteristics were recorded using ICD-10 codes (Appendix A). Age, sex, type of regimen (FOLFOX or FOLFIRINOX), concomitant use of epidermal growth factor receptor (EGFR) inhibitors, immune checkpoint inhibitors (ICIs) and bevacizumab, baseline arterial hypertension, ischemic heart disease, heart failure, chronic kidney disease, cerebrovascular disease, and epilepsy were considered as confounders of the causal pathway between having an early-onset 5-FU toxicity and death, treatment cessation, or hospital admission, and were included in multiple variable models (Appendix A). A confounder is a variable that is associated with both the exposure (early-onset 5-FU toxicity) and the outcome (death, treatment cessation, or hospital admission), potentially distorting the true causal relationship between them, leading to an overestimation or underestimation of the true effect of the exposure on the outcome. Each confounder was selected using a plausible mechanistic approach, considering its potential impact on both the development of 5-FU toxicity and the risk of adverse outcomes. For example, chronic kidney disease could confound the relationship between 5-FU toxicity and hospital admission, as it may independently increase the risk of hospitalization while also affecting the body’s ability to eliminate 5-FU, thereby influencing the appearance of early-onset 5-FU toxicity.

### 2.6. Power Calculation

The study yielded an 80% power to detect a 1% difference in the rate of death at the end of follow-up with a hazard ratio (HR) of 1.19 assuming a sample size of 4000, a 20% incidence of the explanatory variable (early-onset 5-FU toxicities), and a two-sided alpha level of 0.05.

### 2.7. Statistical Methods

Descriptive statistics were used to summarize the demographic characteristics of the patients and relevant clinical information. Continuous variables were presented as median and interquartile range (IQR). Categorical variables were summarized using absolute frequencies and percentages.

Kaplan–Meier curves were constructed using early-onset toxicity occurrence in the first cycle as the explanatory variable to assess overall survival. After checking for Cox proportional hazards assumptions using the Schoenfeld residuals-based method [24], this regression model was used to account for confounding variables previously stated with a two-sided alpha of 0.05.

Sensitivity analyses were conducted by exploring the Kaplan–Meier curves of patients receiving FOLFOX and FOLFIRINOX separately, performing individual log-rank tests.

The chi-squared test or Fisher’s exact test was used to test the association between independent variables and secondary outcomes. A multiple variable logistic regression model was used to assess the association between early-onset toxicities and secondary outcomes accounting for the confounding variables listed previously, with a two-sided alpha of 0.05 considered statistically significant.

All statistical analyses were performed using R version 4.4.0 (R Core Team, 2024) [25].

## 3. Results

### 3.1. Descriptive Statistics

The eligible source population was composed of 7146 patients who had a cancer diagnosis and received at least one dose of 5-FU. After applying the exclusion criteria, 3988 antimetabolite-naïve patients were included for analysis, with 3189 receiving the FOLFOX regimen and 799 FOLFIRINOX (Figure 1). The median age of the cohort was 62.9 years (IQR 54.7–70.4) and 57.5% were male. Colorectal and pancreatic cancers were the most frequent diagnoses and the FOLFIRINOX regimen was administered in 20% of patients in this cohort. Testing for DPD deficiency was performed in 49 patients, of whom 7 tested positive. In total, 1 of 9 patients (11%) with early-onset toxicity and 6 of 40 patients (15%) without early-onset toxicity had some degree of DPD deficiency (Table 1). Early-onset toxicities occurred in 763 (19.1%) patients undergoing the first cycle of FOLFOX or FOLFIRINOX. Vomiting (8.6%), thrombocytopenia (3.1%), and diarrhea (2.9%) had the highest incidence and were more frequent in the patients receiving FOLFIRINOX. (Table 2). The occurrence of cardiovascular toxicities did not differ between patients receiving FOLFOX and FOLFIRINOX (0.4% vs. 0.0%, Fisher’s exact test, *p*-value = 0.14). None of the patients who had early-onset toxicities received uridine triacetate.

### 3.2. Primary Outcome

Patients who experienced early-onset toxicity had a worse prognosis, with a median overall survival of 2.5 years [95% confidence interval {95% CI} 2.2 to 2.9] compared with 5.3 years [95% CI 4.7 to 5.8] in those without an early-onset toxicity (Figure 2, log-rank test, *p* < 0.001). After adjusting for age, sex, type of regimen (FOLFOX or FOLFIRINOX), concomitant use of EGFR inhibitors, ICIs and bevacizumab, baseline arterial hypertension, ischemic heart disease, heart failure, chronic kidney disease, cerebrovascular disease, and epilepsy, the occurrence of early-onset toxicity yielded an HR of 1.61 [95% CI 1.44 to 1.80] in a multiple variable Cox proportional hazards model (*p* < 0.001).

Sensitivity analysis carried out by analyzing the overall survival of patients who only received the FOLFOX regimen revealed a median overall survival of 3.0 years [95% CI 2.5 to 3.9] in patients with early-onset toxicity compared with 6.1 years [95% CI 5.7 to 6.9] in those without early-onset toxicity (Appendix A; log rank test, *p* < 0.001). Patients who received only the FOLFIRINOX regimen had a median overall survival of 1.8 years [95% CI 1.5 to 2.2] in patients with early-onset toxicity compared with 2.3 years [95% CI 2.0 to 2.6] in those without early-onset toxicities (Appendix A; log rank test, *p* < 0.001)

Median survival time was 1.82 years in 2015, decreased to 1.72 years in 2016, and gradually increased to 2.47 years in 2018. After decreasing to 2.25 years in 2019, it reached 1.84 years in 2020, potentially reflecting the impact of the COVID-19 pandemic on cancer care. Median survival time rebounded up to 2.58 years in 2021 and 2.46 years in 2022. Median survival could not be calculated due to insufficient events in 2023; however, trends suggested continued improvement.

### 3.3. Secondary Outcomes

Treatment cessation occurred in 45.0% (1795/3988) of the study population and was more frequent among patients who experienced early-onset toxicities (53.9%, 411/763) compared with those who did not (42.9%, 384/3225) (Chi-square test, *p* < 0.001). After adjusting for age, sex, type of regimen, concomitant use of EGFR inhibitors, ICIs and bevacizumab, baseline arterial hypertension, ischemic heart disease, heart failure, chronic kidney disease, cerebrovascular disease, and epilepsy, the occurrence of early-onset toxicity was significantly associated with early treatment cessation, with an odds ratio (OR) of 1.53 [95% CI 1.30 to 1.80] in a multiple variable logistic regression model (*p* < 0.001).

The frequency of early hospital admission was compared among patients with and without early-onset toxicity. Early admissions occurred in 1.7% (13/763) of the patients who experienced early-onset toxicity compared with 0.19% (6/3225) of those who did not (Fisher’s exact test, *p* < 0.01). Among the 19 patients needing early hospital admission, intensive care unit admission was required in 9 of the 13 patients who had early-onset toxicity compared with 5 of the 6 patients who did not have early-onset toxicity. After adjusting for age, sex, type of regimen, concomitant use of EGFR inhibitors, ICIs and bevacizumab, baseline arterial hypertension, ischemic heart disease, heart failure, chronic kidney disease, cerebrovascular disease, and epilepsy, a complete separation arose when using a multiple variable logistic regression model of the association between early-onset toxicity and early hospital admission. Thus, a Firth’s bias-reduced logistic regression model was used instead, which yielded an OR of 8.69 [95% CI 3.45 to 24.18] (*p* < 0.001) for association of early-onset 5-FU toxicity with unplanned hospitalization.

## 4. Discussion

This study presents the impact of early-onset 5-FU related toxicities experienced during FOLFOX or FOLFIRINOX therapy on long-term survival in patients with cancer within community oncology practices. Data were analyzed from the first cycle of chemotherapy in antimetabolite-naïve cancer patients, as early-onset 5-FU related toxicities often occur during or after the first course of treatment and can be particularly severe [16,17]. Early-onset toxicities were experienced by 19.1% of patients, which was comparable to previous studies reporting rates from 13% up to 30%, and gastrointestinal symptoms (nausea, vomiting) were most frequent [14,16]. The occurrence of early-onset toxicities was associated with a lower median overall survival (2.5 years vs. 5.3 years, *p* < 0.001), a higher likelihood of treatment cessation (53.9% vs. 42.9%, *p* < 0.001), and early hospital admissions (1.70% vs. 0.19%, *p* < 0.001). Multivariate analysis revealed that early-onset toxicities significantly increased the risk of mortality (HR 1.61, *p* < 0.001), treatment cessation (OR 1.53, *p* < 0.001), and early hospital admissions (OR 8.69, *p* < 0.001) after adjusting for confounders. These findings highlight the clinical burden of early-onset toxicities, which not only increase mortality, treatment cessation, and hospital admissions but may also negatively impact patient quality of life and healthcare costs. Interestingly, all types of toxicities were more frequent in those receiving FOLFIRINOX, except for cardiovascular toxicities (Table 2). It is possible that the irinotecan-induced diarrhea, vomiting, and neutropenia might have compounded onto the toxic effects of 5-FU. Moreover, FOLFIRINOX is more widely prescribed for pancreatic cancer, which has a worse prognosis compared with colorectal cancer and renders patients more prone to drug toxicities [7,9].

This study leveraged a large multicenter cohort from the GRN, encompassing 70 community oncology practices across 14 states in the US, which enhanced the generalizability of the findings to real-world settings. The use of an extensive shared EMR system with a comprehensive data warehouse provided a robust and consistent data collection across diverse clinical environments. The inclusion of patients treated in community settings where the majority of patients receive care, adds to the relevance and applicability of the results. The use of real-world data and a causal framework applied to the multivariable adjustments for potential confounders allowed for a more accurate assessment of the impact of early-onset 5-FU-related toxicities on long-term outcomes.

Although early-onset toxicities were experienced in 19% of this cohort, uridine triacetate was not administered to any patient, which suggests that the severity of the reported toxicities might not have been at the level warranting its use. On the other hand, the lack of uridine triacetate delivery in the early stages following 5-FU administration may have contributed to the worse outcomes in the patients who experienced toxicities. The very limited number of patients that received DPD testing reflects the infrequency of its use in the United States, and the data did not provide any information on the type (e.g., targeted genotyping, sequence-based genotyping, DPD enzyme assay) or timing of testing (i.e., prior to chemotherapy or after onset of toxicities) [26,27]. Considering the large variability in non-functional DPYD alleles reported among the different races and ethnicities, other enzyme variations associated with severe 5-FU toxicities (e.g., TYMS, ORPT), and the associated costs, prospective testing may not be feasible in patients receiving 5-FU therapy and may not accurately predict DPD deficiency in all patients tested [1,12,13,15,28]. Uridine triacetate is an FDA-approved antidote for fluoropyrimidine overdose regardless of symptoms, and/or acute severe (grade 3 or greater) toxicities that is provided within 96 h of cessation of fluoropyrimidine delivery, regardless of DPD status [17,18,19]. In community oncology settings, where early recognition of adverse events and rapid diagnostic confirmation of enzyme deficiencies may not be feasible, its use may significantly reduce morbidity and mortality and allow continuation of treatment.

There are several limitations inherent to the retrospective nature of this study that could potentially have led to the underreporting of variables and misclassification bias. The reliance on ICD-10 codes was a significant limitation as it may have led to the underreporting of adverse events, potentially resulting in an incomplete capture of all relevant toxicities and clinical conditions. The severity of toxicities as measured by the common terminology criteria for adverse effects (CTCAE) could not be assessed based on the available data. Detailed information regarding the severity of toxicities would have provided more insight into not only their impact on outcomes but also the decision to prescribe uridine triacetate, which was not provided for any patient. The lack of detailed toxicity severity data limited our ability to assess how varying levels of 5-FU toxicity impact treatment outcomes; patients with mild toxicity may not require treatment, while those with severe reactions may experience life-threatening complications, making it harder to draw firm conclusions about the safety of the treatment. Unmeasured confounding factors, particularly cancer staging, were not accounted for in the analysis as they were not available in the source data; this could influence both the occurrence of early-onset toxicities and patient survival. Moreover, we were unable to assess the relationship between treatment doses and cancer stage due to the lack of detailed staging information in the current dataset. These limitations highlight the need for cautious interpretation of the findings and suggest that further research, potentially involving prospective designs and more comprehensive data collection, is necessary to confirm and extend these results. Future studies should also explore the integration of extensive DPD deficiency data and standardized toxicity grading to improve the understanding of individual susceptibility to early-onset toxicities and guide personalized treatment strategies.

## 5. Conclusions

The occurrence of early-onset 5-FU-related toxicities during the first cycle of FOLFOX or FOLFIRINOX treatment was associated with an increased risk of death, hospitalization, and treatment cessation in antimetabolite-naïve patients receiving care in the community oncology practice setting. A prompt recognition of early-onset toxicities and a timely intervention with effective supportive care practices (e.g., uridine triacetate) would reduce mortality and morbidity, allow for continuation of treatment, and ultimately improve long-term outcomes. Further work is needed to determine the optimal prospective screening for enzyme deficiencies/variations related to severe acute toxicities that can be applied to all patients. In the meantime, the availability of uridine triacetate in community oncology practices can be potentially lifesaving for patients receiving fluoropyrimidine chemotherapy who develop severe acute toxicities.

## Figures and Tables

**Figure 1 cancers-16-04050-f001:**
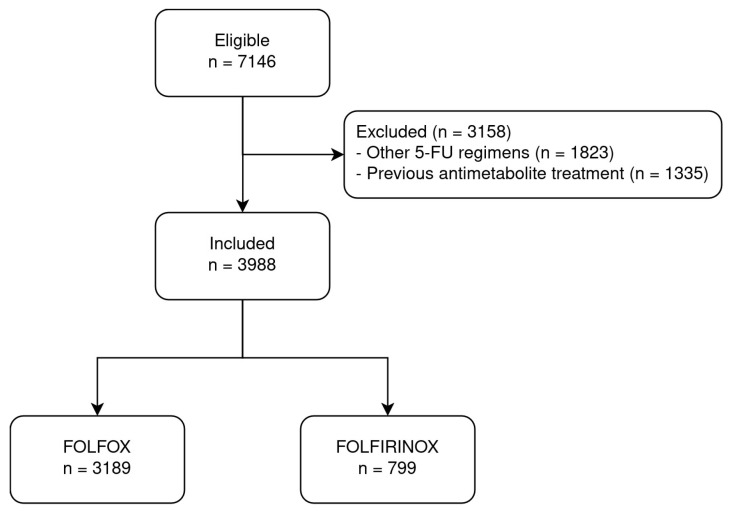
Patient flowchart.

**Figure 2 cancers-16-04050-f002:**
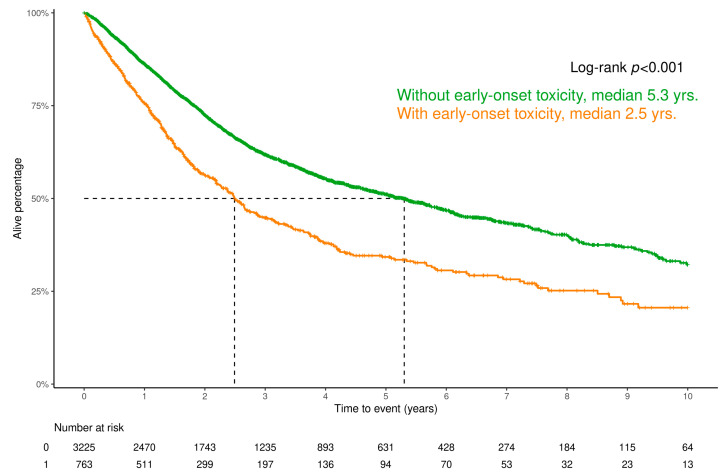
Overall survival by early-onset toxicity occurrence. Patients with early-onset toxicity had a significantly lower median overall survival. After adjustment, early-onset toxicity was associated with a higher hazard ratio (HR 1.61, *p* < 0.001).

**Table 1 cancers-16-04050-t001:** Patient characteristics.

	Total Patients (N = 3988)	With Early-Onset Toxicity (N = 763)	Without Early-Onset Toxicity (N = 3225)
Age, years (interquartile range)	62.9 (54.7–70.4)	61.9 (54.5–69.4)	63.1 (54.7–70.6)
Male, n (%)	2300 (57.7)	386 (50.6)	1914 (59.3)
Cancer diagnosis, n (%)			
Colon-rectum	2497 (63.2)	415 (54.8)	2082 (65.2)
Pancreas	789 (20.0)	201 (26.6)	588 (18.4)
Stomach	248 (6.3)	56 (7.4)	192 (6.0)
Esophagus	241 (6.1)	44 (5.8)	197 (6.2)
Biliary tract	101 (2.6)	20 (2.6)	81 (2.5)
Small intestine	65 (1.6)	15 (2.0)	50 (1.6)
Other	8 (0.2)	6 (0.8)	2 (0.1)
Treatment regimen, n (%)			
FOLFOX	3189 (80.0)	559 (73.3)	2630 (81.6)
FOLFIRINOX	799 (20.0)	204 (26.7)	595 (18.4)
Bevacizumab, n (%)	541 (13.6)	80 (10.5)	461 (14.3)
EGFR inhibitors, n (%)	37 (0.9)	5 (0.7)	32 (1.0)
Immune checkpoint inhibitors, n (%)	83 (2.1)	12 (1.6)	71 (2.2)
Hypertension, n (%)	1616 (40.5)	330 (43.3)	1286 (39.9)
Ischemic heart disease, n (%)	399 (10.0)	72 (9.4)	327 (10.1)
Heart failure, n (%)	132 (3.3)	33 (4.3)	99 (3.1)
Chronic kidney disease, n (%)	227 (5.7)	68 (8.9)	159 (4.9)
Cerebrovascular disease, n (%)	195 (4.9)	39 (5.1)	156 (4.8)
Epilepsy, n (%)	28 (0.7)	11 (1.4)	17 (0.5)
DPD test results, n (%)			
Not tested	3939 (98.8)	754 (98.8)	3185 (98.8)
Tested	49 (1.2)	9 (1.2)	40 (1.2)
Normal/no deficiency	39 (1.0)	5 (0.7)	34 (1.1)
Deficiency NOS	5 (0.1)	1 (0.1)	4 (0.1)
Partial deficiency	2 (0.05)	0	2 (0.05)
Pending results	3 (0.08)	3 (0.4)	0

EGFR—epidermal growth factor receptor; DPD—dihydropyrimidine dehydrogenase; NOS—not otherwise specified.

**Table 2 cancers-16-04050-t002:** Early-onset toxicities.

	Total Patients (N = 3988)	FOLFOX (N = 3189)	FOLFIRINOX (N = 799)
Any toxicity, n (%)	763 (19.1)	559 (17.5)	204 (25.5)
Gastrointestinal, n (%)			
Vomiting	344 (8.6)	258 (8.1)	86 (10.8)
Diarrhea	115 (2.9)	60 (1.9)	55 (6.9)
Mucositis	23 (0.6)	20 (0.6)	3 (0.4)
Hematopoietic, n (%)			
Thrombocytopenia	124 (3.1)	84 (2.6)	40 (5.0)
Neutropenia	19 (0.5)	7 (0.2)	12 (1.5)
Neurological, n (%)	47 (1.2)	33 (1.0)	14 (1.8)
Cardiovascular, n (%)	12.0 (0.3)	12 (0.4)	0 (0.0)

## Data Availability

The data that support the findings of this study are available from the corresponding author upon reasonable request.

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
