# Peer review of "Long-Term Outcomes of 5-Fluorouracil-Related Early-Onset Toxicities: A Retrospective Cohort Study"

_cancers, 2024, doi:10.3390/cancers16234050_

Round 1
Reviewer 1 Report
Comments and Suggestions for Authors
Did the covjd pandemic cause an excess of deaths? The authors make no reference to possibility. Were higher doses of treatment sed i initial round in patients with more further advanced cancers?
Author Response
Dear reviewer,
Thank you for your thoughtful review and constructive feedback. We greatly appreciate the time and effort you have taken to evaluate our manuscript, and we have carefully considered your comments to improve the quality and clarity of our work.
Comment 1: Did the covjd pandemic cause an excess of deaths? The authors make no reference to possibility.
Response 1: We have analyzed the median survival times factored by year of first 5-FU receipt. We found a slight decrease in 2020, given that the following paragraph was added.
Subsection 3.2 Primary outcome, from line 263 to 268
“Median survival time was 1.82 years in 2015, decreased to 1.72 years in 2016, and gradually increased to 2.47 years in 2018. After decreasing to 2.25 years in 2019, it reached 1.84 years in 2020, potentially reflecting the impact of the COVID-19 pandemic on cancer care. Median survival time rebounded up to 2.58 years in 2021 and 2.46 years in 2022. Median survival could not be calculated in 2023 due to insufficient events, however, trends suggested continued improvement.”
Comment 2: Were higher doses of treatment sed i initial round in patients with more further advanced cancers?
Response 2: As mentioned in the Discussion section, the current state of data curation did not allow assessment of cancer stage of each patient.Given that limitation no dose-stage analysis could be done. The following sentence has been added to the Discussion section
Section 4 Discussion, from line 367 to 369:
“Moreover, we were unable to assess the relationship between treatment doses and cancer stage due to the lack of detailed staging information in the current dataset.”

Reviewer 2 Report
Comments and Suggestions for Authors
Interesting work covering a large population of people. Very valuable because of the population data i.e. real data.
I have 2 questions for the researchers:
1- how is it defined as an aside: ischemic heart disease? Because 5FU causes chest pain already during the infusion and there are often changes in the ekg then. This is usually caused by spasm of the coronary artery
Please quote the guidelines:
A practical approach to the 2022 ESC cardio-oncology guidelines: Comments by a team of experts - cardiologists and oncologists.
Leszek P, Klotzka A, Bartuś S, Burchardt P, Czarnecka AM, Długosz-Danecka M, Gierlotka M, Koseła-Paterczyk H, Krawczyk-Ożóg A, Kubiatowski T, Kurzyna M, Maciejczyk A, Mitkowski P, Prejbisz A, Rutkowski P, Sierko E, Sterliński M, Szmit S, Szwiec M, Tajstra M, Tycińska A, Witkowski A, Wojakowski W, Cybulska-Stopa B.
Kardiol Pol. 2023;81(10):1047-1063. doi: 10.33963/v.kp.96840. Epub 2023 Sep 3.PMID: 37660389
2- How do the researchers explain that there were fewer cardiac complications in the FOLFIRNOX group? Since there is an add-on bolus of 5FU?
Author Response
Dear reviewer,
Thank you for your thoughtful review and constructive feedback. We greatly appreciate the time and effort you have taken to evaluate our manuscript, and we have carefully considered your comments to improve the quality and clarity of our work.
Reviewer 2
Interesting work covering a large population of people. Very valuable because of the population data i.e. real data.
I have 2 questions for the researchers:
Comment 1: how is it defined as an aside: ischemic heart disease? Because 5FU causes chest pain already during the infusion and there are often changes in the ekg then. This is usually caused by spasm of the coronary artery
Please quote the guidelines:
A practical approach to the 2022 ESC cardio-oncology guidelines: Comments by a team of experts - cardiologists and oncologists.
Leszek P, Klotzka A, Bartuś S, Burchardt P, Czarnecka AM, Długosz-Danecka M, Gierlotka M, Koseła-Paterczyk H, Krawczyk-Ożóg A, Kubiatowski T, Kurzyna M, Maciejczyk A, Mitkowski P, Prejbisz A, Rutkowski P, Sierko E, Sterliński M, Szmit S, Szwiec M, Tajstra M, Tycińska A, Witkowski A, Wojakowski W, Cybulska-Stopa B.
Response 1: As stated in the methods section all adverse events were imputed using a combination of ICD-10 codes, laboratory results and procedures. In the case of acute ischemic heart disease, ICD-10 codes Angina pectoris (I20), Acute myocardial infarction (I21), Subsequent myocardial infarction (I22), Certain current complications following acute myocardial infarction (I23) and Other acute ischemic heart diseases (I24) AND/OR any request of a troponin test AND/OR a procedure such as cardiac catheterization, coronary angiography, heart angioplasty, cardiac stent placement, or percutaneous coronary intervention. We didn't have at our disposal the EKG results given the unstructured nature of that data.
To clarify this we have added the following sentence:
Subsection 2.3 Explanatory variable, lines 121 to 124:
“...and categorized as gastrointestinal (vomiting, diarrhea, mucositis), hematopoietic (thrombocytopenia, neutropenia), cardiovascular, and neurological using specific criteria for each based on…”.
Also, the tables in the supplementary materials have been updated to match the names of the mapped toxicities as shown in the main manuscript. Specifically, Ischemic heart disease was changed to Cardiovascular and Acute neurological event to Neurological.
Comment 2: How do the researchers explain that there were fewer cardiac complications in the FOLFIRNOX group? Since there is an add-on bolus of 5FU?
Response 2: Although the counts of cardiovascular adverse events differ between the groups, it is essential to consider the relative percentages: 0.4% for FOLFOX and 0.0% for FOLFIRINOX. A Fisher's Exact Test yields a p-value of 0.141, indicating that the difference is not statistically significant. We have added the following sentence to clarify this:
Subsection 3.1 Descriptive statistics, lines 198 to 199:
“The occurrence of cardiovascular toxicities did not differ between FOLFOX and FOLFIRINOX patients (0.4% vs 0.0%, Fisher’s exact test, p-value = 0.14)”

Reviewer 3 Report
Comments and Suggestions for Authors
Q1: The flow from the introduction through the methodology to results is generally well-organized. The study design and analysis appear robust, with thorough explanations of statistical methods and controls for confounding factors.
Q2: Study Design: This section is well-detailed, though you could improve clarity by breaking it into smaller subsections (e.g., "Study Design and Setting," "Patient Population").
Q3: Consider placing more emphasis on the study’s aims earlier, particularly why early-onset toxicity in community settings requires investigation.
Q4: While the need for uridine triacetate is mentioned, highlighting it as a critical intervention based on this study’s data might give more weight to the recommendations.
Q5: The tables are informative but could be accompanied by brief summaries to aid understanding, especially for complex variables like DPD deficiency.
Q6: Confounders and Statistical Models: The choice of confounders is well-justified, but adding a brief rationale for each confounder might strengthen this section.
Q7: You mention limitations regarding ICD-10 codes, retrospective design, and lack of toxicity severity data. Expanding slightly on each limitation with specific examples or scenarios where these factors may affect results would provide a balanced perspective.
Q8: Figure 2: This figure provides a clear comparison of survival, but additional labeling or a brief figure caption summarizing the primary finding would make it immediately interpretable without referring back to the text.
Q9: The section on treatment cessation and hospital admissions is clear but could benefit from a more prominent focus on the clinical significance of these findings. For instance, you could emphasize that unplanned admissions impact patient quality of life and healthcare costs.
Q10: In the discussion, the narrative could be restructured slightly to focus more immediately on a concise summary of the study’s main findings and implications before moving into broader comparisons with other studies.
Q11: Highlight the significance of findings on early hospital admission rates more prominently in the discussion, as it supports the study's practical recommendations.
Q12: In the conclusions, it might help to briefly restate the importance of timely intervention with uridine triacetate, as it is a key recommendation based on the findings.
Q13: Including a few more actionable future research directions would strengthen this section. For example, suggest prospective studies to further evaluate toxicity and the timing of DPD testing.
Q14: The conclusion reiterates the importance of early recognition and intervention but could benefit from a more specific takeaway on how this impacts patient management in real-world settings.
Q15: conclusion: You may consider emphasizing that proactive monitoring in community practices, with protocols for rapid intervention, could mitigate risks for patients with high susceptibility to 5-FU toxicity.
Q16: Please pay attention to all abbreviations (e.g., EGFR, ORPT) , which are defined at first use.
Author Response
Dear reviewer,
Thank you for your thoughtful review and constructive feedback. We greatly appreciate the time and effort you have taken to evaluate our manuscript, and we have carefully considered your comments to improve the quality and clarity of our work.
Reviewer 3
Comment 1: The flow from the introduction through the methodology to results is generally well-organized. The study design and analysis appear robust, with thorough explanations of statistical methods and controls for confounding factors.
Response 1: Thank you for the comment, we highly appreciate it.
Comment 2: Study Design: This section is well-detailed, though you could improve clarity by breaking it into smaller subsections (e.g., "Study Design and Setting," "Patient Population").
Response 2: Thank you for the comment. Nevertheless, we think that the current structure of the methods section is specific enough and would lose consistency if we divide it into smaller sections.
Comment 3: Consider placing more emphasis on the study’s aims earlier, particularly why early-onset toxicity in community settings requires investigation.
Response 3: We believe that the rationale for investigating early-onset toxicities in the community practice setting was presented in the third paragraph of the introduction (“The incidence, severity, and clinical impact of 5-FU toxicities have not been well documented in community oncology practices where 85% of oncology patients in the United States receive their care. Patients in community practices are comprised of diverse socioeconomics, races, and ethnicities and present with various comorbidities, all of which may be associated with unusual patterns of toxicities and management, and thus provide real-world data on treatment outcomes”). The aim of the study was placed accordingly at the end of this paragraph after providing the aforementioned rationale for investigating this topic. We have clarified the aim as follows:
Section 1 Introduction, lines 95 to 99:
“This study was undertaken to investigate the association between 5-FU related early-onset toxicities and long-term survival outcomes in cancer patients in community oncology practices. Specifically, to ensure toxicities were attributable to 5-FU, this study focused solely on antimetabolite-naïve patients who had received their first cycle of first-line FOLFOX or FOLFIRINOX therapy.”
Comment 4: While the need for uridine triacetate is mentioned, highlighting it as a critical intervention based on this study’s data might give more weight to the recommendations.
Response 4: We had introduced uridine triacetate in the second paragraph of the introduction in the context of its therapeutic impact in cases of severe 5-FU toxicities or overdose, and the necessity of its availability as testing for enzyme deficiencies are not widespread in lines 77-85.
Comment 5: The tables are informative but could be accompanied by brief summaries to aid understanding, especially for complex variables like DPD deficiency.
Response 5: A brief summary of the relevant data can be found in the text. For example, “Testing for DPD deficiency was performed in 49 patients, of whom 7 tested positive. One of 9 patients (11%) with early-onset toxicity and 6 of 40 patients (15%) without early-onset toxicity had some degree of DPD deficiency (Table 1).”. This summary should suffice as we aimed to minimize redundant information between the tables and the text, ensuring a clear and concise presentation of the data without unnecessary repetition. Further, a line item was added and rows were reformatted under DPD test results to better delineate the patients who received testing from those who did not.
Comment 6: Confounders and Statistical Models: The choice of confounders is well-justified, but adding a brief rationale for each confounder might strengthen this section.
Response 6: Thank you for the comments, we have added the following sentences given that all the potential confounding variables were selected using the same approach:
Subsection 2.5 Diagnosis, baseline characteristics and potential confounders, lines to 146 to 156:
“A confounder is a variable that is associated with both the exposure (early-onset 5-FU toxicity) and the outcome (death, treatment cessation, or hospital admission), potentially distorting the true causal relationship between them, leading to an overestimate or underestimate of the true effect of the exposure on the outcome. Each confounder was selected using a plausible mechanistic approach, considering its potential impact on both the development of 5-FU toxicity and the risk of adverse outcomes. For example, chronic kidney disease could confound the relationship between 5-FU toxicity and hospital admission, as it may independently increase the risk of hospitalization while also affecting the body's ability to eliminate 5-FU, thereby influencing the appearance of early-onset 5-FU toxicity.”
Comment 7: You mention limitations regarding ICD-10 codes, retrospective design, and lack of toxicity severity data. Expanding slightly on each limitation with specific examples or scenarios where these factors may affect results would provide a balanced perspective.
Response 7: We appreciate the comment, nevertheless we have already made a detailed explanation about each limitation on the last paragraph of the discussion section. We have added the following sentence for clarification purposes:
Section 4. Discussion, lines 360 to 365:
“The lack of detailed toxicity severity data limits our ability to assess how varying levels of 5-FU toxicity impact treatment outcomes; patients with mild toxicity may not require treatment, while those with severe reactions may experience life-threatening complications, making it harder to draw firm conclusions about the safety of the treatment.”
Comment 8: Figure 2: This figure provides a clear comparison of survival, but additional labeling or a brief figure caption summarizing the primary finding would make it immediately interpretable without referring back to the text.
Response 8: We have added the following caption to the figure
Lines 270 to 272:
“Patients with early-onset toxicity had a significantly lower median overall survival. After adjustment, early-onset toxicity was associated with a higher hazard ratio (HR 1.61, p<0.001).”
Comment 9: The section on treatment cessation and hospital admissions is clear but could benefit from a more prominent focus on the clinical significance of these findings. For instance, you could emphasize that unplanned admissions impact patient quality of life and healthcare costs.
Response 9: Thank you for your suggestion. While we acknowledge the clinical significance of unplanned hospital admissions and their potential impact on patient quality of life and healthcare costs, unfortunately, we did not have access to data on these aspects in our study. We have expanded our discussion on this topic:
Section 4. Discussion, lines 325 to 338:
“The occurrence of early-onset toxicities was associated with a lower median overall survival (2.5 years vs. 5.3 years, p<0.001), a higher likelihood of treatment cessation (53.9% vs. 42.9%, p<0.001) and early hospital admissions (1.70% vs 0.19%, p<0.001). Multivariate analysis revealed that early-onset toxicities significantly increased the risk of mortality (HR 1.61, p<0.001), treatment cessation (OR 1.53, p<0.001) and early hospital admissions (OR 8.69, p<0.001) after adjusting for confounders. These findings highlight the clinical burden of early-onset toxicities, which not only increase mortality, treatment cessation and hospital admissions but may also negatively impact patient quality of life and healthcare costs.”
Comment 10: In the discussion, the narrative could be restructured slightly to focus more immediately on a concise summary of the study’s main findings and implications before moving into broader comparisons with other studies.
Response 10: We have reorganized the discussion to prioritize clarity and focus, moving the main findings to the opening paragraph. Previously, these findings were presented in the second paragraph.
Comment 11: Highlight the significance of findings on early hospital admission rates more prominently in the discussion, as it supports the study's practical recommendations.
Response 11: We have added the following:
Section 4. Discussion, lines 325 to 338:
“The occurrence of early-onset toxicities was associated with a lower median overall survival (2.5 years vs. 5.3 years, p<0.001), a higher likelihood of treatment cessation (53.9% vs. 42.9%, p<0.001) and early hospital admissions (1.70% vs 0.19%, p<0.001). Multivariate analysis revealed that early-onset toxicities significantly increased the risk of mortality (HR 1.61, p<0.001), treatment cessation (OR 1.53, p<0.001) and early hospital admissions (OR 8.69, p<0.001) after adjusting for confounders. These findings highlight the clinical burden of early-onset toxicities, which not only increase mortality, treatment cessation and hospital admissions but may also negatively impact patient quality of life and healthcare costs.”
Comment 12: In the conclusions, it might help to briefly restate the importance of timely intervention with uridine triacetate, as it is a key recommendation based on the findings.
Response 12: We believe that the importance of timely use of uridine triacetate is appropriately stated in the third paragraph of the discussion section
“Although early-onset toxicities were experienced in 19% of this cohort, uridine triacetate was not administered to any patient, which suggests that the severity of the reported toxicities might not have been at the level warranting its use. On the other hand, the lack of uridine triacetate delivery in the early stages following 5-FU administration may have contributed to the worse outcomes in the patients who experienced toxicities. The very limited number of patients that received DPD testing reflects the infrequency of its use in the United States, and the data did not provide any information on the type (e.g., targeted genotyping, sequence-based genotyping, DPD enzyme assay) or timing of testing (i.e., prior to chemotherapy or after onset of toxicities) [26,27]. Considering the large variability in non-functional DPYD alleles reported among the different races and ethnicities, other enzyme variations associated with severe 5-FU toxicities (e.g., TYMS, ORPT), and the associated costs, prospective testing may not be feasible in patients receiving 5-FU therapy and may not accurately predict DPD deficiency in all patients tested [1,12,13,15,28]. Uridine triacetate is an FDA-approved antidote for fluoropyrimidine overdose regardless of symptoms, and/or acute severe (grade 3 or greater) toxicities that is provided within 96 hours of cessation of fluoropyrimidine delivery, regardless of DPD status [17-19]. In community oncology settings, where early recognition of adverse events and rapid diagnostic confirmation of enzyme deficiencies may not be feasible, its use may significantly reduce morbidity and mortality as well as allow continuation of treatment.”
Comment 13: Including a few more actionable future research directions would strengthen this section. For example, suggest prospective studies to further evaluate toxicity and the timing of DPD testing.
Response 13: We have added the following sentence:
Section 4. Discussion, lines 372 to 375:
“Future studies should also explore the integration of extensive DPD deficiency data and standardized toxicity grading to improve the understanding of individual susceptibility to early-onset toxicities and guide personalized treatment strategies.”
Comment 14: The conclusion reiterates the importance of early recognition and intervention but could benefit from a more specific takeaway on how this impacts patient management in real-world settings.
Response 14: We believe that the sentence stated the following sentence reflects our view on the impact of the findings “Prompt recognition of early-onset toxicities and timely intervention with effective supportive care practices (e.g., uridine triacetate) would reduce mortality and morbidity, allow for continuation of treatment, and ultimately improve long-term outcomes.”
Comment 15: conclusion: You may consider emphasizing that proactive monitoring in community practices, with protocols for rapid intervention, could mitigate risks for patients with high susceptibility to 5-FU toxicity.
Response 15: The limited data on DPD-deficient patients restricted our ability to draw definitive conclusions, as we were unable to evaluate their heightened susceptibility to 5-FU toxicity within the scope of our dataset.
Comment 16: Please pay attention to all abbreviations (e.g., EGFR, ORPT) , which are defined at first use.
Response 16: We have thoroughly reviewed the manuscript to ensure consistency in language and the appropriate use of abbreviations.

Round 2
Reviewer 3 Report
Comments and Suggestions for Authors
accepted